ⓐ | **Open Peer Review** | Bacteriology | Research Article

# Multi-omics analyses reveal fecal microbial community and metabolic alterations in finishing cattle fed probiotics-fermented distiller's grains diets

Rong Zhang,[1,2] Shihui Mei,[1,2] Guangxia He,[1,2] Miaozhan Wei,[1,2] Lan Chen,[1,2] Ze Chen,[1,2] Min Zhu,[1,2] Bijun Zhou,[1,2] Kaigong Wang,[1,2] Zhentao Cheng,[1,2] Chunmei Wang,[1] Erpeng Zhu,[1,2] Chao Chen[1]

**ABSTRACT**    Distiller's grains (DG) are a potential source of animal feeds, and many studies have indicated positive regulatory roles of feeding DG diets in animal breeding. However, there is currently a dearth of research on the actions and underlying mechanisms of probiotics-fermented distiller's grains (FDG)-based diets in cattle breeding. This study aimed to assess the impact of integrating FDG into the diet of finishing cattle on their fecal microbial community and metabolites. Thirty Simmental crossbred cattle (local yellow cattle × Simmental cattle, 8.5 months old, 420.38 ± 68.11 kg) were selected and randomly divided into three dietary treatments, including the basal diet group (CON group), the FDG replacing 10% concentrate (FDG-10%) group, and the FDG replacing 20% concentrate (FDG-20%) group. 16S and ITS sequencing of fecal samples collected from each group on the 30th day of the formal feeding suggested that feeding FDG diets had little effect on the composition and diversity of fecal bacterial and fungal communities in finishing cattle. However, the relative abundance of cellulose-degrading bacteria, including the *Christensenellaceae R-7 group* and *Ruminococcaceae family* was significantly higher in both the FDG-20% vs CON comparison and the FDG-20% vs FDG-10% comparison. Besides, the FDG-10% group had a significant drop in the relative abundance of *Aspergillus* and a noteworthy increase in the relative abundance of *Candida* when compared to the CON group. Non-targeted metabolomics analysis showed that the addition of FDG modified the levels of organoheterocyclic compounds, lipids and lipid-like molecules, and benzenoids in the feces of finishing cattle and significantly enhanced the metabolic pathway of bile secretion. Further correlation analyses suggested a close association between the significantly differential fecal microbiota and metabolites. In conclusion, these results suggest that FDG supplementation has little effect on the structure and diversity of the fecal microbiota in finishing cattle, but alters intestinal metabolite profiles and influences bile secretion pathways by modulating the relative abundance of genera of fecal bacteria and fungi *Christensenellaceae R-7 group*, *Lachnospiraceae_NK3A20_group*, *Mucor*, and *Candida*. These findings provide a scientific theoretical basis for the use of FDG in animal feeds.

**IMPORTANCE**  Probiotics-fermented distiller's grains (FDG) are potential feed sources for livestock. Here, we investigated the effects of partially replacing concentrates with FDG on fecal bacterial and fungal community structure and metabolic profiles in finishing cattle. The results reveal that feeding FDG-based diets alters intestinal metabolite profiles and up-regulates bile secretion pathways through the regulation of relative abundance of certain fecal genera. These findings provide some new insights into clarifying the role and potential mechanisms of FDG diets and also offer a scientific basis for the development of FDG into functional feed resources.

**Peer Reviewers** Adetomiwa Ayodele Adeniji, Stellenbosch University, Cape Town, Western Cape, South Africa; Ritesh Kumar Aggarwal, Indian Institute of Technology, New Delhi, Delhi, India; Yuchao Zhao, Beijing University of Agriculture, Beijing, China

Address correspondence to Erpeng Zhu, zhu13782701756@126.com, or Chao Chen, chenc@gzu.edu.cn.

The authors declare no conflict of interest.

**KEYWORDS** probiotics-fermented distiller's grains, feces, metabolomics, fecal microbiota, ITS

With the rapid development of the global livestock industry, issues of feed shortages and increasing prices have arisen. To produce safe, inexpensive, and nutritious feed has become one of the most urgent issues in livestock husbandry. Distiller's grains (DG), a by-product of the wine industry, are favored because of their low production cost and high nutritional value. Studies have shown that DG has good effects as an alternative feed for different animal species (1–5). One of the primary uses of DG is to replace a certain proportion of concentrates to feed livestock, including direct feeding and feeding after drying or fermentation treatment, of which microbial fermentation of DG has a wider range of applications and gives better results. Microbial fermentation mainly adopts probiotic fermentation. Probiotics-fermented distiller's grains (hereafter collectively referred to as fermented distiller's grains, FDG) are mainly utilized due to the fact that probiotics can use the sugars and starch present in DG to ferment and produce a variety of nutrients such as proteins, cellulase, fatty acids, etc. (6). Studies have also shown that FDG can improve the quality and palatability of DG, inhibit the proliferation of harmful bacteria, and maintain the micro-ecological balance of the intestinal tract in Simmental crossbred cattle and broilers (7, 8). Therefore, FDG has become one of the hot topics of research into the feed utilization of DG.

The development of high-throughput sequencing technology, high-resolution mass spectrometry, and data integration and analysis technology has fostered new break-throughs in systems biology research characterized by multi-omics (9). Multi-omics aims to integrate genomics, epigenomics, transcriptomics, proteomics, and metabolomics in an unbiased way to systematically resolve complex mechanisms and phenotypes of animal life systems. Intestinal microbes and their interactions with the host play a crucial role in the health of the host organism (10). Multi-omics can be used to systematically study the biology of the host intestine and to characterize host-intestinal microbiome interactions using deeply integrated technologies, and this can reveal the complex regulatory mechanisms associated with the growth and development of animals, as well as the mechanisms underlying the development and treatment of various diseases (11–13). In recent years, omics analysis has also been used to study the effects of FDG diets on animals. Our previous studies have suggested that feeding dried distillers' grains and FDG fermented by commercial microbial preparations altered the microbial community structure and metabolic patterns of the gastrointestinal tract in Guanling cattle and Guanling crossbred cattle, and that replacing part of the concentrates with FDG fermented by commercial microbial preparations also altered the amino acid composition and proportions in beef, thus improving its flavor in Guanling cattle (14–18). To further investigate the effects of FDG diets on fecal microbial structure and metabolic profiles, here we selected four widely used probiotics (*Enterococcus faecalis*, *Lactobacillus plantarum*, *Aspergillus niger*, and *Saccharomyces cerevisiae*) to supplement FDG and then partially replaced the concentrates of basal diets to feed finishing cattle. Finally, 16S and ITS high-throughput sequencing and liquid chromatography-mass spectrometry (LC-MS) metabolomics were performed to investigate the effects of feeding FDG diets on the structure of the fecal microbiota and the metabolome of finishing cattle, thus screening out the key flora, metabolites, and metabolic pathways as potential biomarkers, which will provide references for the feasibility of FDG as an alternative animal feed and shed new light on the mitigation of the shortage of feed resources in the livestock industry.

## RESULTS

### Feeding FDG diets maintains the fecal bacterial community

Our analysis revealed no significant change in average daily gain (ADG) of finishing cattle in all groups after feeding FDG diets (Table S2). We investigated the changes in bacterial communities of rectal feces in finishing cattle following FDG feeding by 16S high-throughput sequencing. Operational taxonomic units (OTUs) are taxonomic unit

markers in phylogenetic and population genetics studies. Generally, sequences with 97% similarity were assigned to the same OTUs, which means a microbial species or genus. The diversity and the abundance of different microbes in a test sample are based on the analysis of OTUs. A Venn diagram showed that we identified 2,380 common OTUs in all samples analyzed, with 1,670, 1,655, and 1,439 OTUs specific to the CON, FDG-10%, and FDG-20% group, respectively (Fig. 1A). The dilution curves generated for each group of OTUs showed that the identification rate of OTUs eventually stabilized as the number of sample reads increased, suggesting that the current sequencing depth and sample size are sufficient to assess the major members of the fecal bacterial community (Fig. 1B). The results of the alpha diversity analysis showed that Chao1, ACE, Shannon, and Simpson indices in the FDG-10% and FDG-20% groups had no statistically significant differences in comparison with those in the CON group ($P > 0.05$) (Fig. 1C). This indicated that feeding FDG diets does not have a significant effect on fecal bacterial diversity and richness in finishing cattle. To compare the distribution of fecal microbiota in different groups, beta diversity analysis was performed by principal coordinate analysis (PCoA). The results showed that the microbial communities could not be clearly separated in the CON, FDG-10%, and FDG-20% groups (Fig. 1D). These results indicated that the fecal bacterial communities were relatively stable.

Analysis of the bacterial community composition of the fecal microbiota showed that the dominant phyla in the CON, FDG-10%, and FDG-20% groups were *Bacteroidetes* and *Firmicutes* at the phylum level (Fig. 2A). At the genus level, *Ruminococcaceae.UCG.005* and *Rikenellaceae.RC9.gut.group* were the dominant group of bacteria across all groups (Fig.

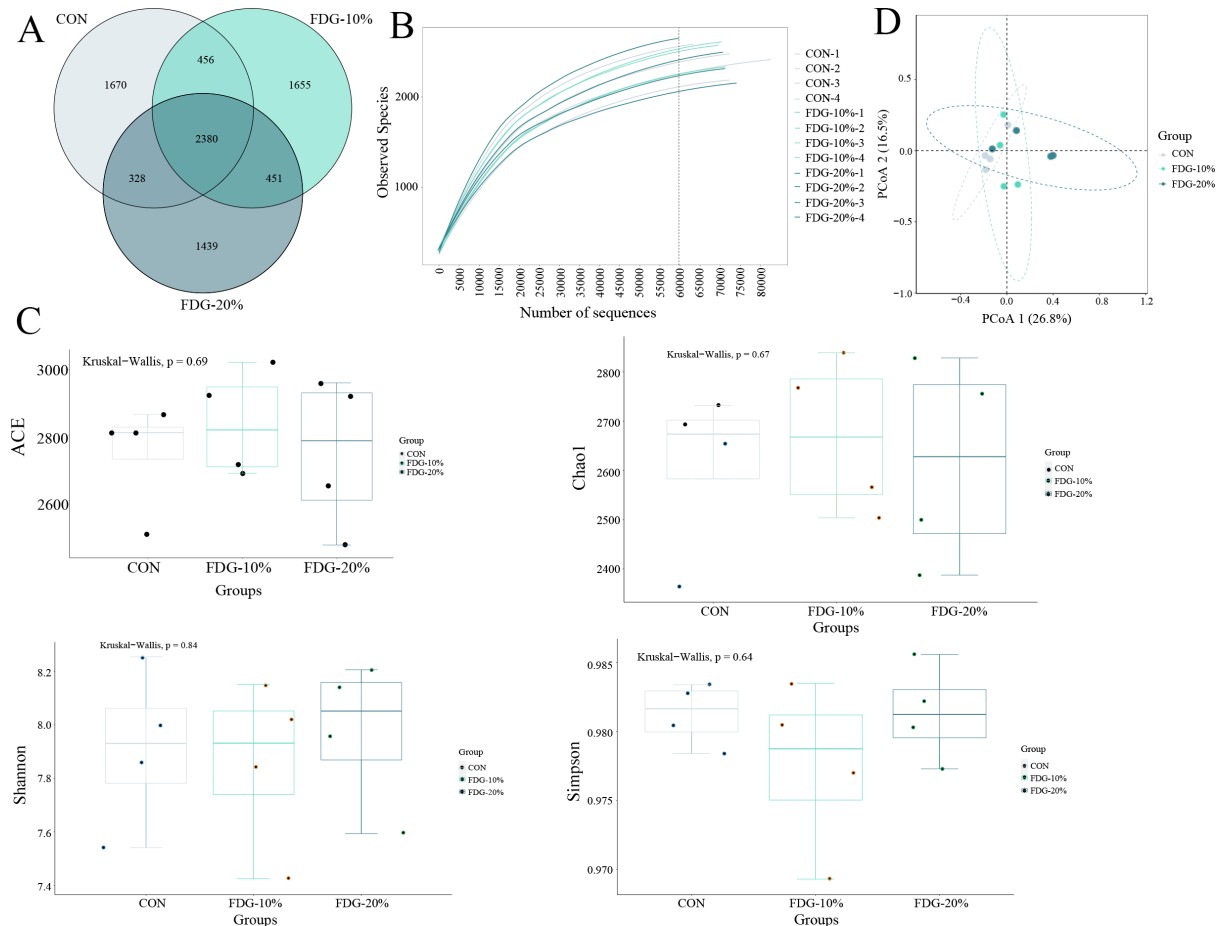

FIG 1 Bacterial diversity across the three groups is relatively stable. (A) Venn diagram illustrating the OTUs in three groups. (B) Rarefaction curve. (C) Alpha diversity analysis. (D) Principal-coordinate analysis. The basal diet group, the FDG replacing 10% concentrate group, and the FDG replacing 20% concentrate group are denoted as CON, FDG-10%, and FDG-20%, respectively. Each experimental group comprises fecal samples randomly collected from four cattle ($n = 4$).

2B). Linear discriminant analysis effect size (LEfSe) analysis demonstrated a statistically significant increase in the relative abundance of *Ruminococcaceae UCG-014* in the FDG-10% group compared to the CON group (*P* < 0.05) (Fig. 2C). The FDG-20% group displayed a significantly higher relative abundance of *Family XIII AD3011 group*, *Christensenellaceae R-7 group*, *Lachnospiraceae_NK3A20_group*, and *Ruminococcaceae NK4A214 group* (*P* < 0.05), and a significantly lower relative abundance of *Bacteroidales_RF16 group* (*P* < 0.05) when compared to the CON group (Fig. 2D). Besides, the FDG-20% group exhibited a markedly higher (*P* < 0.05) relative abundance of *Family XIII AD3011 group* and *Lachnospiraceae_NK3A20_group* compared to the FDG-10% group. Conversely, the relative abundance of *Bacteroidales_RF16_group* in the FDG-20% group showed a significant decrease compared to that in the FDG-10% group (*P* < 0.05) (Fig. 2E). The results also showed that, compared to the CON and FDG-10% groups, the FDG-20% group displayed a significant increase in the relative abundance of both the *Family XIII AD3011 group* (*P* < 0.05) and the *Lachnospiraceae_NK3A20_group* (*P* < 0.05), and a significant decrease in the relative abundance of the *Bacteroidales_RF16_group* (*P* < 0.05).

## Feeding FDG diets stabilizes the richness of fecal fungal communities

We further investigated the changes in fungal communities of rectal feces in finishing cattle following FDG feeding by ITS high-throughput sequencing. ITS sequencing results showed that clustering according to the 97% sequence similarity criterion produced 2,081 OTUs, with 81 OTUs being shared by all groups (Fig. 3A). Dilution curves, produced for each group of OTUs, indicated that the current sequencing depth and sample size are adequate to assess the primary components of the fecal fungal community

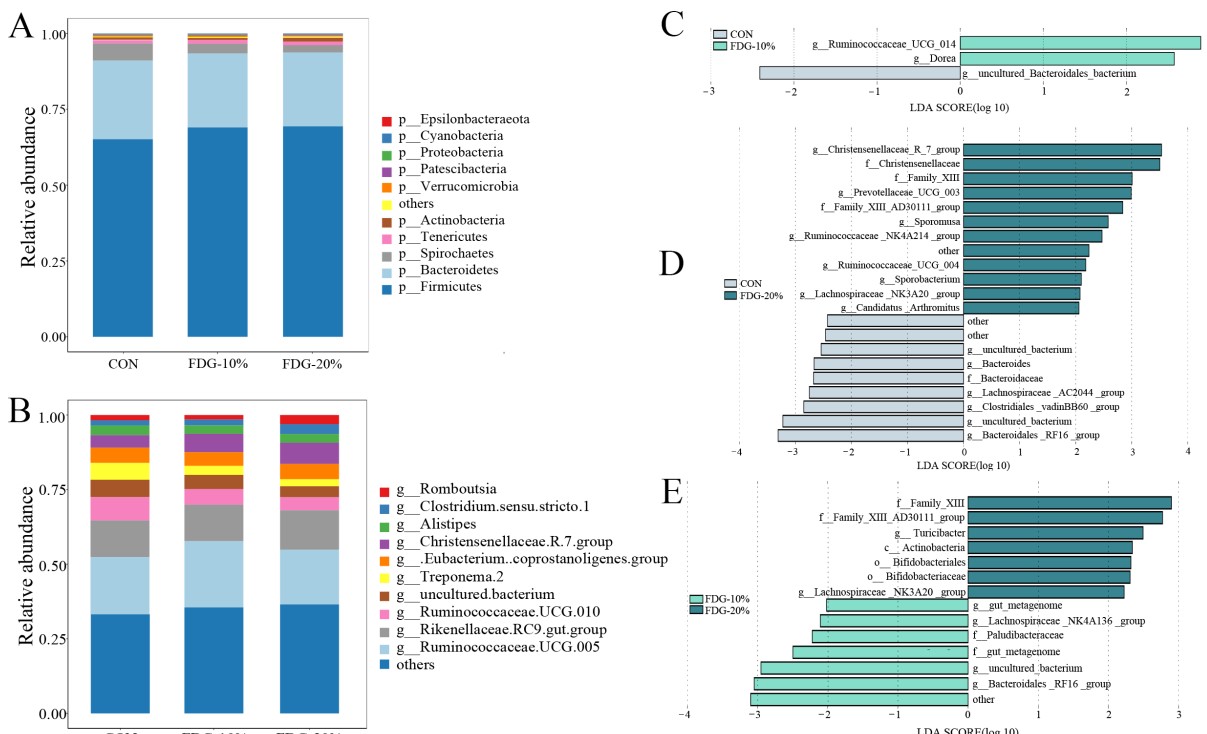

**FIG 2** The dominant phyla were *Bacteroidetes* and *Firmicutes*, and at the genus level, *Ruminococcaceae.UCG.005* and *Rikenellaceae.RC9.gut.group* were the dominant group. (A) Proportions of bacterial communities by phylum. (B) Proportions of bacterial communities by genus. (C–E) The LEfSe analyses indicate differences in intestinal bacterial community composition between groups, with a linear discriminant analysis (LDA) score of >2 and a *P*-value of <0.05. The basal diet group, the FDG replacing 10% concentrate group, and the FDG replacing 20% concentrate group are denoted as CON, FDG-10%, and FDG-20%, respectively. Each experimental group comprises fecal samples randomly collected from four cattle (*n* = 4). Abbreviations used were p for phylum, c for class, o for order, f for family, and g for genus.

(Fig. 3B). The alpha diversity analysis results indicated that there were no significant statistical differences in the Chao1 and ACE indices of the FDG-10% and FDG-20% groups compared to the CON group ($P > 0.05$). However, the Shannon and Simpson indices were found to be notably higher in the FDG-10% group than in the CON group ($P < 0.05$) (Fig. 3C). These results revealed that feeding FDG diets doesn't significantly affect the richness of fecal fungal communities in finishing cattle. However, feeding 10% FDG diets notably increases the diversity of fungal communities. Furthermore, PCoA analysis showed a clear distinction in fecal fungal composition between the CON, FDG-10%, and FDG-20% groups, which served as a more effective grouping effect (Fig. 3D).

At the phylum level, *Ascomycota* reigned supreme in all groups (Fig. 4A). *Talaromyces* and *Aspergillus* were the main genera identified in all groups (Fig. 4B). However, the relative abundance of *Ascomycota* was significantly lower, and the relative abundance of *Basidiomycota* was significantly higher both in the FDG-20% vs CON comparison and the FDG-20% vs FDG-10% comparison (Fig. 4C through E). Compared to the CON group, the LEfse analysis revealed a significant decrease in the relative abundance of *Aspergillus* in the FDG-10% group ($P < 0.05$), while *Candida* showed a significant increase in the same group ($P < 0.05$) (Fig. 4C). Furthermore, *Sporisorium* exhibited a significant increase in the relative abundance in the FDG-20% group ($P < 0.05$), while the relative abundance of *Issatchenkia* and *Wickerhamomyces* was significantly lower in the same group ($P < 0.05$) (Fig. 4D). The FDG-20% group had a higher relative abundance of *Aspergillus*, whereas the relative abundances of *Issatchenkia* and *Candida* were significantly lower in comparison to the FDG-10% group (Fig. 4E).

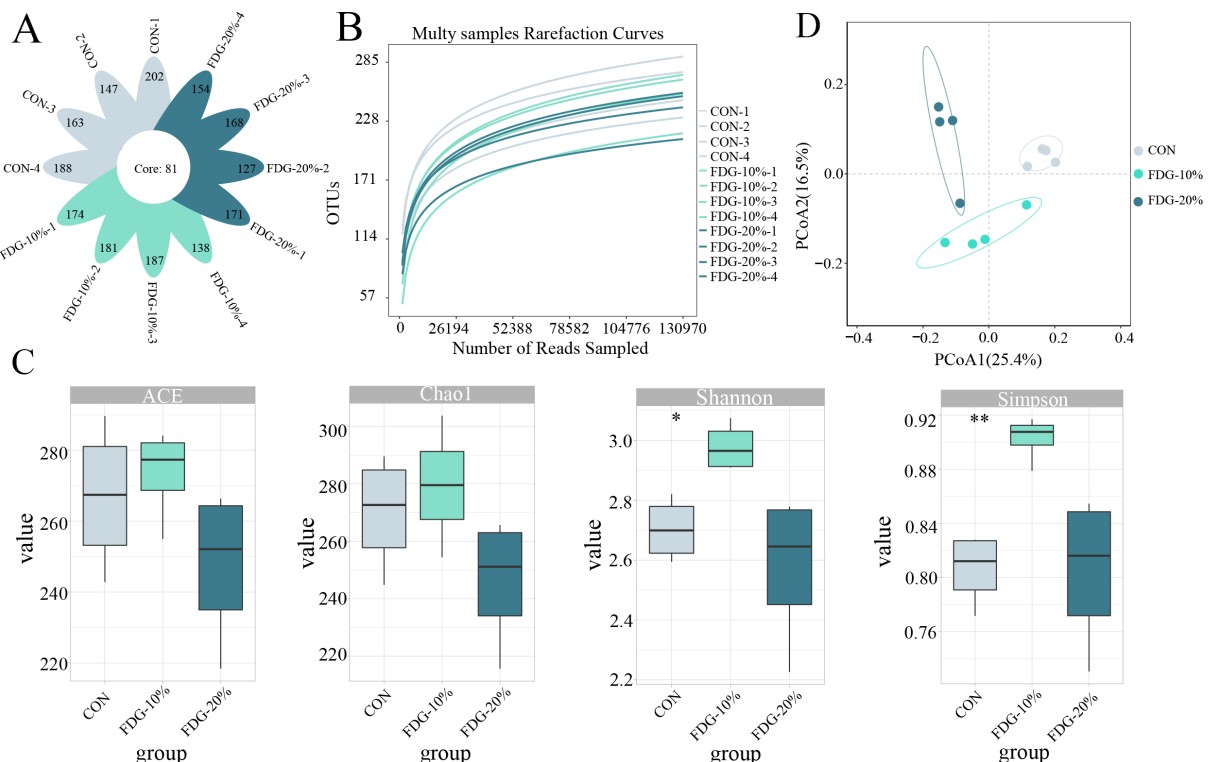

**FIG 3** The richness of intestinal fungal communities is relatively stable between groups. (A) Venn diagram illustrating the OTUs in three groups. (B) Rarefaction curve. (C) Alpha diversity analysis. (D) Principal-coordinate analysis. The basal diet group, the FDG replacing 10% concentrate group, and the FDG replacing 20% concentrate group are denoted as CON, FDG-10%, and FDG-20%, respectively. Each experimental group comprises fecal samples randomly collected from four cattle ($n = 4$).

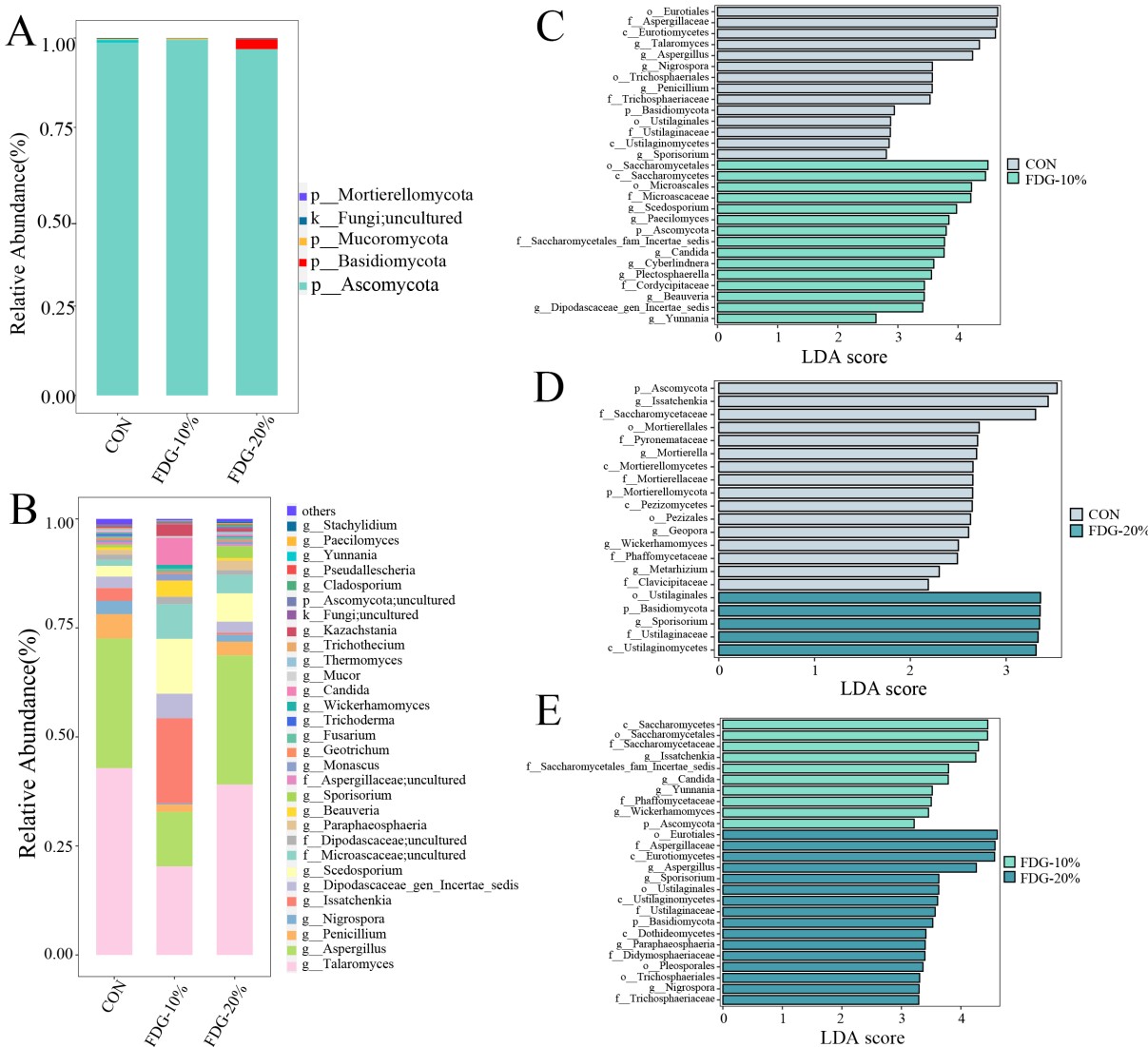

**FIG 4** At the phylum level, *Ascomycota* reigns supreme in all groups, and *Talaromyces* and *Aspergillus* are the main genera identified in all groups. (A) Proportions of fungal communities by phylum. (B) Proportions of fungal communities by genus. (C–E) The LEfSe analyses indicate differences in intestinal fungal community composition between groups, with an LDA score of >2 and a *P*-value of <0.05. The basal diet group, the FDG replacing 10% concentrate group, and the FDG replacing 20% concentrate group are denoted as CON, FDG-10%, and FDG-20%, respectively. Each experimental group comprises fecal samples randomly collected from four cattle (*n* = 4). Abbreviations used were p for phylum, c for class, o for order, f for family, and g for genus.

## Feeding FDG diets alters intestinal metabolism in finishing cattle

After investigating the effects of feeding DFG diets on the gut microbial community in finishing cattle, we further used non-targeted metabolomic analysis to examine rectal feces from finishing cattle to explore the effect of feeding FDG diets on the metabolism of intestinal flora. We analyzed intestinal flora metabolites and found 1,206 and 803 metabolites in the positive and negative ion modes, respectively. Principal component analysis (PCA) and partial least squares discriminant analysis (PLS-DA) were used to depict group differences. The greater the separation between the sample groups in the plots, the more significant the difference is. The score plots for PCA and PLS-DA demonstrated a distinct variation between the CON, FDG-10%, and FDG-20% groups, suggesting a significant separation effect among these groups (Fig. 5).

After analysis by LC-MS, the metabolites with significant differences were screened out based on the partial least squares discriminant analysis (OPLS-DA) model with

variable influence on projection (VIP) >1 and $P < 0.05$ as criteria. The results were illustrated by volcano plots, which showed that compared with the CON group, there were 221 differential metabolites in the FDG-10% group, including 187 up-regulated and 34 down-regulated differential metabolites; compared with the CON group, there were 320 differential metabolites in the FDG-20% group, of which 262 were up-regulated differential metabolites and 58 were down-regulated differential metabolites. Additionally, compared with the FDG-10% group, there were 236 differential metabolites in the FDG-20% group, including 218 up-regulated and 18 down-regulated differential metabolites (Fig. 6A through C). Table 1 presents information on the top 10 significant difference metabolites in the FDG-10% vs CON comparison, the FDG-20% vs CON comparison, and the FDG-20% vs FDG-10% comparison in positive ion mode. Information on all the differential metabolites obtained in this study is listed in Table S2. These differential metabolites predominantly comprised organoheterocyclic compounds, organic oxygen compounds, organic acids and derivatives, lipids and lipid-like molecules, benzenoids, lignans, neolignans, and related compounds (Fig. 6D through F).

Furthermore, we utilized the Kyoto Encyclopedia of Genes and Genomes (KEGG) database to enrich the metabolic pathways of the differential metabolites. This approach can facilitate a deeper comprehension of the impact of FDG diets on intestinal metabolites in finishing cattle. It was found that there was a significant difference in the metabolic pathway related to bile secretion between the FDG-10% group and the CON group (Fig. 7A). Furthermore, five metabolic pathways exhibited significant differences in the FDG-20% group vs CON comparison, which included taste transduction, GABAergic synapse, bile secretion, retrograde endocannabinoid signaling, and synaptic vesicle cycle (Fig. 7B). There were no metabolic pathways that differed significantly in the FDG-20% group vs FDG-10% comparison. These results above indicate that feeding both 10% and

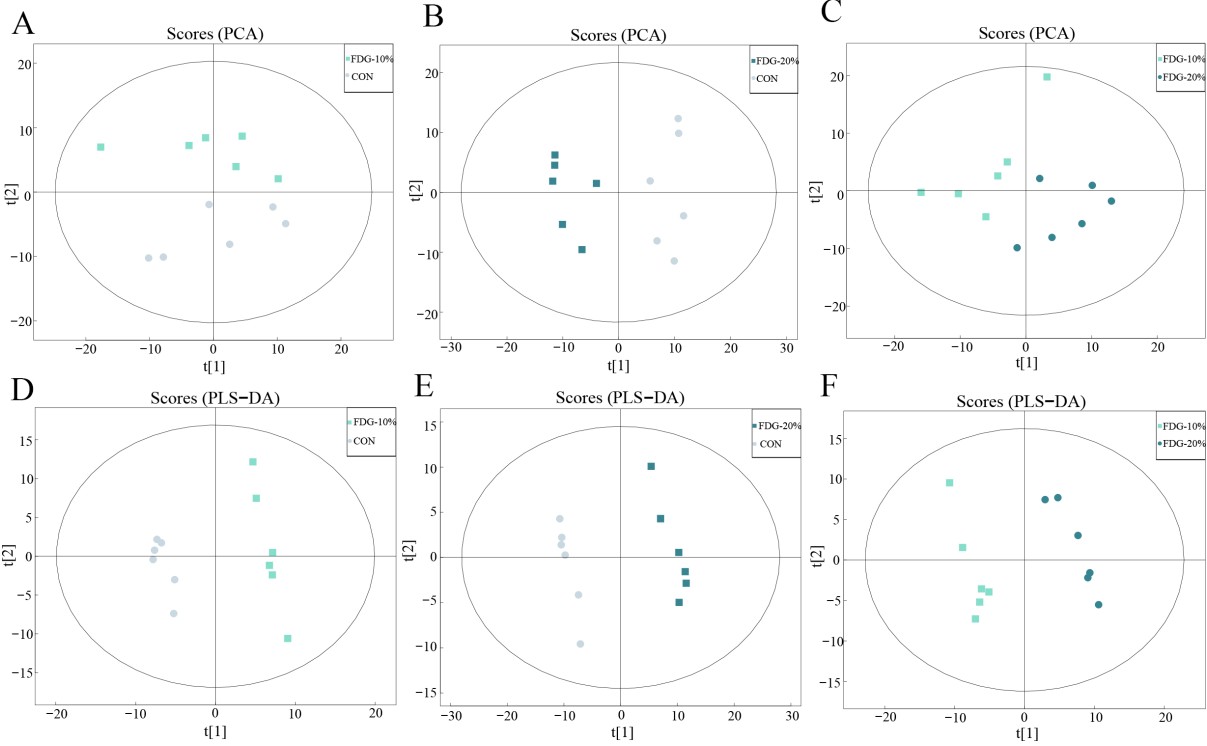

**FIG 5** Feeding FDG diets alters intestinal metabolic patterns in finishing cattle. (A–C) Plot of PCA scores in the positive ion mode in the FDG-10% group vs CON group, FDG-20% group vs CON group, and FDG-20% group vs FDG-10% group comparisons, respectively. (D–F) Plot of PLS-DA scores in the positive ion mode in the FDG-10% group vs CON group, FDG-20% group vs CON group, and FDG-20% group vs FDG-10% group comparisons, respectively. Each experimental group comprises fecal samples randomly collected from six cattle ($n = 6$).

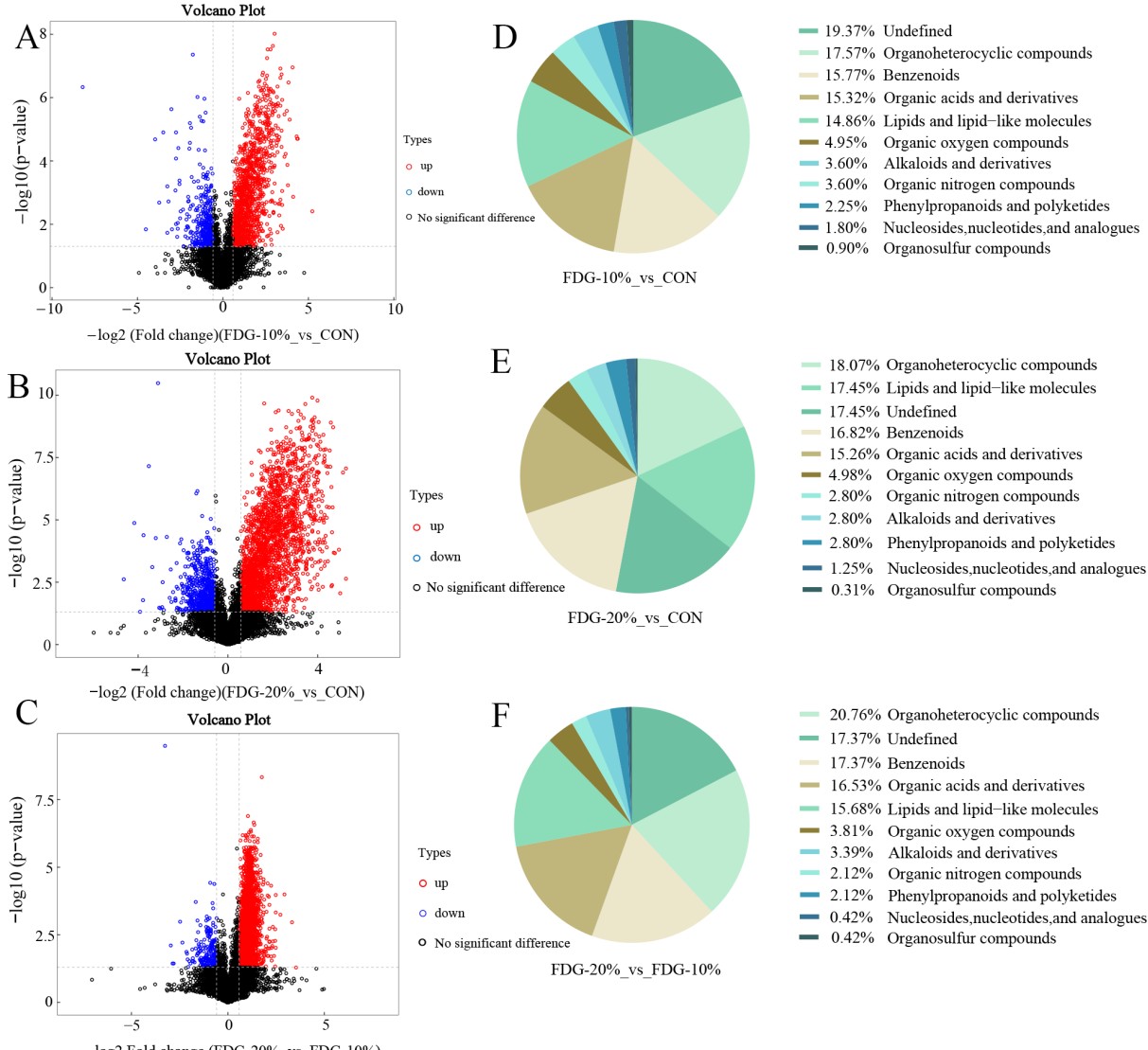

**FIG 6** Feeding FDG diets up-regulates intestinal metabolite levels in finishing cattle. (A–C) Volcano plot of differential metabolites in positive ion mode in the FDG-10% group vs CON group, FDG-20% group vs CON group, and FDG-20% group vs FDG-10% group comparisons, respectively. (D–F) Classification map of differential metabolite sub-classes in the positive ion mode in the FDG-10% group vs CON group, FDG-20% group vs CON group, and FDG-20% group vs FDG-10% group comparisons, respectively. CON, FDG-10%, and FDG-20% for the basal diet group, the FDG replacing 10% concentrate group, and the FDG replacing 20% concentrate group, respectively. Each experimental group comprises fecal samples randomly collected from six cattle ($n = 6$).

20% FDG diets had a positive effect on bile secretion in the intestines of finishing cattle. Additionally, feeding a 20% FDG diet mainly had a positive effect on taste transduction in finishing cattle.

## Close relationship between the differential bacterial and fungal genera and differential metabolites in finishing cattle fed FDG diets

To reveal the association between fecal microbiome and differential metabolites induced by FDG diets, we investigated the potential co-occurrence between the top 10 differential metabolites of positive and negative ions according to VIP values with differential bacterial and fungal genera, respectively, using correlation analyses based on Spearman statistics. Information on the correlation analysis of complete differential metabolites with differential bacterial and fungal genera is listed in Fig. S1. Compared to the CON group, the differential bacterial genera *Ruminococcaceae_UCG_014*

**TABLE 1** The significantly differential metabolites in positive ion modes (TOP 10)[a]

| Metabolites | log$_2$(FC) | P | VIP |
| --- | --- | --- | --- |
| FDG-10% vs CON | | | |
| Sinapine | 20.652 | 0.000 | 20.280 |
| Silodosin | 7.379 | 0.000 | 14.082 |
| .gamma.-glu-cys | 20.165 | 0.000 | 13.734 |
| N-acetyl-d-galactosamine | 0.531 | 0.047 | 9.864 |
| D-glucosaminic acid | 2.077 | 0.004 | 9.630 |
| Dethiobiotin | 6.210 | 0.000 | 9.526 |
| Citalopram | 4.242 | 0.000 | 8.330 |
| 2-Chloro-2',6'-diethylacetanilide | 0.508 | 0.001 | 8.168 |
| Celaxanthin | 0.714 | 0.002 | 7.215 |
| Lenalidomide | 3.832 | 0.001 | 7.064 |
| FDG-20% vs CON | | | |
| Sinapine | 27.054 | 0.000 | 16.819 |
| Silodosin | 12.223 | 0.000 | 13.636 |
| D-glucosaminic acid | 3.894 | 0.000 | 11.908 |
| Dethiobiotin | 15.742 | 0.000 | 11.880 |
| Citalopram | 11.812 | 0.000 | 11.502 |
| .gamma.-glu-cys | 26.383 | 0.000 | 11.400 |
| DL-tyrosine | 7.589 | 0.000 | 8.478 |
| 4,4'-Diaminodiphenylmethane | 19.684 | 0.000 | 8.252 |
| Lenalidomide | 6.582 | 0.000 | 7.603 |
| Vasicinone | 12.950 | 0.000 | 7.140 |
| FDG-20% vs FDG-10% | | | |
| Citalopram | 0.359 | 0.005 | 10.937 |
| Dethiobiotin | 0.394 | 0.000 | 10.807 |
| D-glucosaminic acid | 0.533 | 0.000 | 9.980 |
| Silodosin | 0.604 | 0.005 | 8.666 |
| 4,4'-Diaminodiphenylmethane | 0.387 | 0.000 | 7.500 |
| DL-tyrosine | 0.464 | 0.000 | 7.464 |
| Olomoucine | 0.471 | 0.000 | 7.042 |
| Benzamide, n-[2-[[(3r)-1-[trans-4-hydroxy-4-(6-methoxy-3-pyridinyl) cyclohexyl]-3-pyrrolidinyl]amino]-2-oxoethyl]-3-(trifluoromethyl)- | 0.757 | 0.025 | 6.658 |
| Vasicinone | 0.404 | 0.000 | 6.139 |
| Artemisinin | 0.425 | 0.000 | 6.007 |

[a]VIP, variable influence on projection; FC, fold change; CON, FDG-10%, and FDG-20% for the basal diet group, the FDG replacing 10% concentrate group, and the FDG replacing 20% concentrate group, respectively.

and *Dorea* in the FDG-10% group were positively correlated with the differential metabolites including 3-hydroxybiphenyl, Enterolactone_neg, .gamma.-glu-cys, Biopterin_neg, Cidofovir_neg, and were negatively correlated with 5-aminosalicylic acid_neg, 2-chloro-2',6'-diethylacetanilide_pos; the differential bacterial genera *Prevotellaceae_UCG_003*, *Candidatus_Arthromitus*, *Family_XIII_AD3011_group*, *Ruminococcaceae_NK4A214_group*, *Christensenellaceae_R_7_group*, *Ruminococcaceae_UCG_004*, and *Sporobacterium* in the FDG-20% group were all positively correlated with 3-hydroxyphenylacetic acid_neg, .gamma.-glu-cys_pos, 16.alpha.-hydroxyestrone_pos, and all were negatively correlated with 3-hydroxyphenylacetic acid_neg, .gamma.-glu-cys_pos, 16.alpha. acid_neg, .gamma.-aminobutyric acid_pos (Fig. 8A and B). Compared to the CON group, the differential fungal genera *Scedosporium*, *Mucor*, *Dipodascaceae_gen_Incertae_sedis*, and *Candida* in the FDG-10% group were all positively correlated with 3-hydroxykynurenine_neg, Enterolactone_neg, Biopterin_neg, Cidofovir_neg, .gamma.-glu-cys_pos, 17alpha-ethynylestradiol_pos,

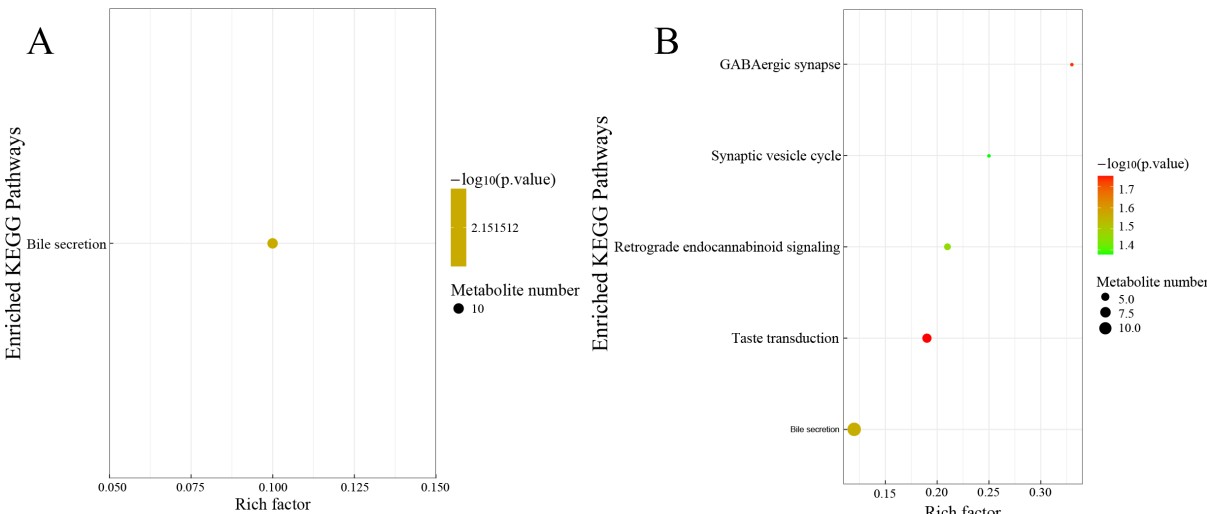

**FIG 7** Feeding FDG diets enhances intestinal bile acid metabolism. (A) KEGG pathway analysis in the FDG-10% vs CON comparison. (B) KEGG pathway analysis in the FDG-20% vs CON comparison. The X-axis represents pathway impact and the Y-axis represents the pathway enrichment. The larger size of the circle indicates greater pathway enrichment, and the darker color indicates higher pathway impact values. The closer the color is to red, the smaller the *P*-value.

16.alpha.-hydroxyestrone_pos and were negatively correlated with .gamma. amino-butyric acid_pos, 2-chloro-2',6'-diethylacetanilide_pos, 5-aminosalicylic acid_neg, and Ile-Pro_neg; and the differential fungal genera *Anaeromyces* in the FDG-20% group was positively correlated with 3-hydroxykynurenine_neg, .gamma.-glu-cys_pos, 16.alpha.-hydroxyestrone_pos, while with .gamma.-aminobutyric acid_pos, 3-hydroxyphenylacetic acid_neg were negatively correlated (Fig. 8C and D). These findings indicate a close relationship between the fecal microbiota and their metabolites in finishing cattle fed with FDG diets.

## DISCUSSION

As an unconventional feed resource, DG possesses beneficial feeding qualities because of its high yield and rich nutrients, and an increasing number of studies have demonstrated the potential of DG as animal feed (19). A large number of bacteria, fungi, archaea, and protozoa inhabit the intestinal tract of ruminants, collectively forming a complex micro-ecosystem, which plays significant roles in intestinal and organic health. Our study aimed to investigate the impact of feeding probiotics-FDG on the structure of fecal microbiota in finishing cattle via high-throughput sequencing of 16S and ITS. The results revealed that supplementation of FDG did not result in notable alterations to the ACE, Chao1, Shannon, and Simpson indicators of the fecal bacterial population in finishing cattle. This suggests that FDG had no significant influence on the structural diversity of the fecal bacterial community of finishing cattle, which is in line with previous studies (20). The species composition of fecal microbes was analyzed further, revealing that the dominant bacterial communities present at the phylum level were *Bacteroidetes* and *Firmicutes*. This is consistent with previous findings on the intestinal microbiota of ruminants (21). Studies have demonstrated that the *Family XIII AD3011* category improves animals' disease resistance, and the *Christensenel-laceae_R-7_group* regulates animal immunocompetence and maintains gastrointestinal homeostasis (22, 23). The prevailing bacterial species in the *Mycobacterium* phylum is *Prevotellaceae UCG-003*, which is predominantly engaged in the catabolism of starch, proteins, xylan, and pectin (24, 25). The *Lachnospiraceae_NK3A20_group* is mainly involved in carbohydrate metabolism. Research has shown that an increase in concentrate intake results in a considerable increase in this group (26). Our study found that the relative abundance of the genera *Family XIII AD3011 group*, *Prevotellaceae UCG-003*, and *Lachnospiraceae_NK3A20_group* was significantly higher in the FDG-20% group,

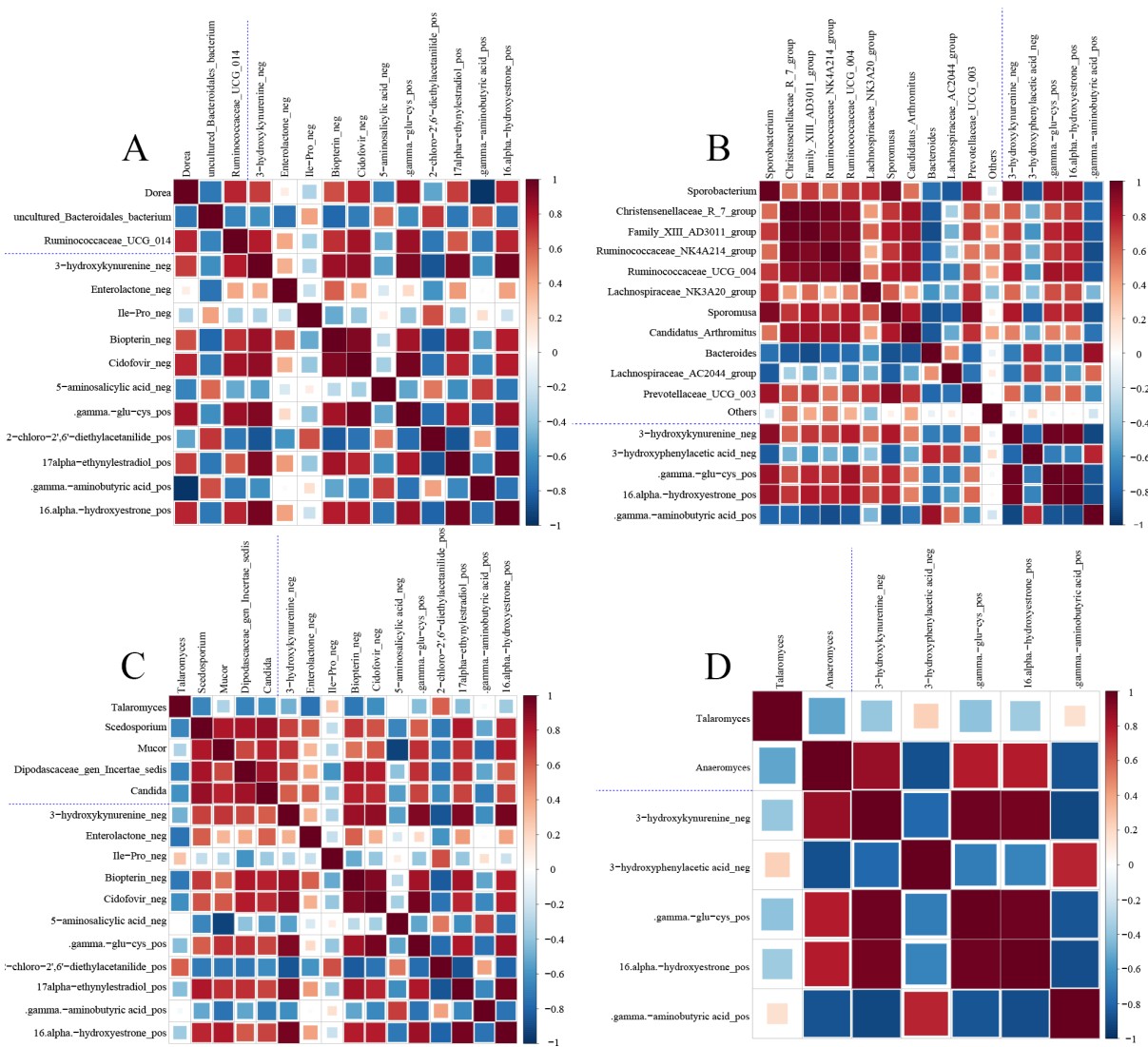

**FIG 8** There is a correlation between the intestinal microbiota and their metabolites in finishing cattle. (A and B) Correlation analysis between significantly differential intestinal bacterial communities and significantly the top 10 differential metabolites of positive and negative metabolites in the FDG-10% vs CON group and FDG-20% vs CON group. (C and D) Correlation analysis between significantly differential intestinal fungal communities and significantly differential top 10 differential metabolites of positive and negative metabolites in the FDG-10% vs CON and FDG-20% vs CON groups. Red and blue colors indicate positive and negative correlations, respectively, and the color scale indicates the magnitude of the correlation coefficient.

which is beneficial for improving the immunity and resistance to diseases of fattening cows, as well as decomposing nutrients in the food to provide nutrition to fattening cattle (27). It is worth noting that the *Christensenellaceae_R-7_group* is also considered a cellulolytic bacterium, similar to the *Ruminococcaceae family* (28). The FDG-20% group significantly increased the relative abundance of cellulose-degrading bacteria, such as *Christensenellaceae R-7 group* and *Ruminococcaceae family*, which digested fibers from fermented lees into nutrients that are readily absorbed by cattle. The research above indicates that FDG has little impact on the structure and diversity of bacterial flora in finishing cattle's gastrointestinal tracts. However, substituting 20% of concentrate with FDG alters the proportional abundance of certain fiber-degrading bacteria, such as *Christensenellaceae R-7 group* and *Ruminococcaceae family*, which are associated with carbohydrate metabolism and the breakdown of cellulose within FDG.

The ITS sequencing results demonstrated that feeding FDG diets had no significant effect on the abundance of fecal fungi in finishing cattle. However, feeding 10% FDG

substantially increased the diversity of the fecal fungal community. This study revealed that the *Ascomycetes* were the dominant phylum of fungi, and the *Talaromyces* and *Aspergillus* were their dominant genera. Several studies have indicated that over 98% of known fungi belong to the phylum *Ascomycetes* and *Aspergillus* (29), which is also in agreement with our findings. Both *Cyberlindnera* and *Candida* are yeasts, and studies have shown that *Candida* has a positive effect on institutional productivity, health, and immune performance (30). Yeast or yeast products can be used as a protein source and have been shown to promote growth performance, modulate the intestinal microbiota, and positively affect the immune system (31, 32). *Paecilomyces* has various biological activities such as an anti-microbial, anti-tumor, insecticidal, and anti-malarial parasite, and *Paecilomyces variotii* can be fermented to produce a range of extracellular enzymes including cellulase, xylanase, amylase, and pectinase (33, 34). The FDG-10% group markedly decreased the relative abundance of *Aspergillus* while significantly increasing that of *Candida*, *Paecilomyces,* and *Cyberlindnera*, and provided proteins and various extracellular enzymes that are beneficial to the immunity and health of finishing cattle. *Aspergillus* is a widely distributed group of filamentous fungi that degrade cellulose, hemicellulose, and lignin. This genus is also the primary fungal genus in saucy baijiu spirits. Its prevalence and ability to decompose facilitate the efficient recycling of organic waste in various ecological niches (35). Feeding FDG at a 10% level resulted in a decrease in the relative abundance of *Aspergillus*. However, the relative abundance of this genus returned to normal when the proportion of FDG was increased to 20%, which may be related to the amount of FDG added. The aforementioned findings demonstrate that the administration of 20% FDG had a minimal effect on the configuration and diversity of the gastrointestinal fungal microorganisms in finishing cattle. Conversely, the supplementation of 10% FDG considerably augmented the population of the said microorganisms while modifying the proportional distribution of fecal fungal genera such as *Aspergillus*, *Candida*, *Paecilomyces, Cyberlindnera* and others, which play an integral role in decomposing fermented distiller's grains and nourishing finishing cattle.

Intestinal microorganisms serve various metabolic functions, including the production of various metabolites. Therefore, fecal metabolites were analyzed in this study. The results revealed the screening of 221 and 320 differential metabolites under positive ion mode in the two treatment groups compared to the control group. There were 236 differential metabolites identified in the FDG-20% group in comparison with the FDG-10% group, and the majority of these exhibited a trend toward down-regulation. Notably, Quinine and Serotonin, along with other differential metabolites, showed significant up-regulation. Quinine is a bitter compound that activates bitter taste receptors (Tas2rs), which are primarily responsible for the perception of bitter flavor and serve to prevent organisms from ingesting toxic bitter substances (36, 37). Quinine has been found to have antimalarial properties (38, 39), and it also assists in eliminating bacterial hosts by impeding the interaction between hemoglobin and dedicatorofcytokinesis8 (DOCK8) (40). Serotonin was significantly upregulated in the experimental group, with the metabolic pathways involved being taste transduction and bile secretion. Enterochromaffin cells in the gastrointestinal tract are the primary source of 5-HT production, with certain intestinal bacteria also producing 5-hydroxytryptamine. In addition to its function as a neurotransmitter, 5-HT exerts a regulatory influence on immune cell activation (41). We discovered a close association between these metabolites and fecal microbiota genera that exhibited marked changes in their relative abundance, including *Prevotellaceae_UCG_003*, *Ruminococcaceae_NK4A214_group*, *Christensenellaceae_R_7_group*, and *Lachnospiraceae_NK3A20_group*, *Mucor*, and *Candida,* based on Spearman's correlation analysis. Consequently, these results suggest that feeding FDG diets regulates the proportional abundance of the genus of fecal bacteria and fungi *Christensenellaceae R-7 group, Lachnospiraceae_NK3A20_group, Mucor, Candida* and, as a result, alters metabolite levels.

To uncover metabolic variations between experimental and control groups, KEGG pathway analysis was conducted on the differential metabolites. Bile secretion is a vital producer of bile acids, which serve as important regulators of life activities and are involved in glucose, fat, and energy metabolism. Additionally, bile acids are closely tied to intestinal hormones, microbiota, and energy balance (42). In the intestine, certain bile acids undergo complex biotransformation, typically facilitated by bacterial enzymes present in the intestinal tract (43). Intestinal flora, including *Bacteroidetes* spp., *Bifidobacterium* spp., and *Lactobacillus* spp., have been found to be involved in bile acid metabolism (44). Bile acids play an important role in determining the abundance and diversity of the microbiota. An imbalance in bacterial bile acid conversion may contribute to metabolic, inflammatory, infectious, and neoplastic diseases (45). Taste is a fundamental physiological sense for organisms that is necessary for nutritional balance. Oral and gastrointestinal sensory systems contain the majority of taste receptors (46). The gastrointestinal system can detect nutrients and toxins through comparable taste receptors and signaling, with enteroendocrine cells shown to be the primary chemosensory cells of the tract (47). In the intestine, the enteroendocrine system comprises the largest endocrine organ in the human body, producing numerous peptide hormones (48, 49). Feeding FDG diets had an effect on metabolite levels in finishing cattle, as indicated by the differential metabolites that were significantly enriched in the bile secretion pathway in both experimental groups and in the taste transduction pathway in the FDG-20% group. This suggests that FDG has a primary effect on bile secretion in finishing cattle, and that the addition of 20% FDG also has an effect on taste transduction in these animals. This was also demonstrated in our previous study. However, further research is needed to determine how FDG affects the proportion of certain fecal bacteria and fungi to regulate bile secretion and taste transduction in finishing cattle.

## Conclusion

Collectively, the present results in this study indicate that feeding FDG diets had little effect on the structure or diversity of the fecal flora. However, it did modify metabolite levels by altering the relative abundance of certain genera of fecal bacteria and fungi. Additionally, the supplementation of FDG regulates bile secretion and taste transduction metabolic pathways, potentially affecting energy balance and taste in finishing cattle, in which the fecal flora *Bacteroidetes* spp., *Bifidobacterium* spp., and *Lactobacillus* spp. may play an important role. Furthermore, the substitution of FDG for 20% of the concentrate resulted in an increased relative abundance of certain cellulolytic bacteria (*Christensenellaceae R-7 group* and *Ruminococcaceae family*), suggesting that this may be a more optimal proportion for FDG substitution. However, further comprehensive research is required to fully substantiate this conclusion. These findings not only offer new insights into the regulatory roles of feeding FDG diets in fecal microbial community structures and metabolites in finishing cattle but also to some degree contribute to the understanding of the mechanisms of action of FDG in cattle husbandry.

## MATERIALS AND METHODS

### Preparation of probiotics-fermented distiller's grains

The DG used in this study was obtained from the Kweichow Moutai Group, Moutai Town, Renhuai City, Guizhou Province, China, whose main components are distilled sorghum and wheat. DG was then fermented with four probiotics including *Enterococcus faecalis*, *Lactobacillus plantarum*, *Aspergillus niger*, and *Saccharomyces cerevisiae. Saccharomyces cerevisiae* and *Enterococcus faecalis* were isolated, identified, and preserved by the Institute of Animal Disease, Guizhou University (Guizhou, China), and *Lactobacillus plantarum* (ACCC11095) and *Aspergillus niger* (CICC2377) were obtained from the Shanghai Bioresource Collection Center (Shanghai, China). Baking soda was added to the DG at 3.5% (m/m), and then the DG was mixed with cornflour, wheat

bran, and rapeseed meal in the ratio of 77%:6%:5%:12% (m/m), and finally, the probiotic solution (mixture of the above-mentioned four probiotics prepared in the ratio of 1:1:1:1 at the concentration of $1 \times 10^8$ CFU/mL) was added to the FDG in the ratio of 8% (vol/m), and then the solution was mixed well. After mixing, the FDG was placed in tonne bags and sealed to ferment for 5 days and then fed as indicated.

## Animals, diet, and experimental design

This experiment was conducted from August to December 2022 at the beef cattle breeding base in Ao Tian Village, Plain Town, Dejiang County, Guizhou Province (108°28′E, 28°N). The trial consisted of a 15-day adaptation period and a 30-day formal feeding period. During the 15-day adopting period, 30 healthy 8.5-month-old Simmental crossbred steers (420.38 ± 68.11 kg) were randomly divided into three groups, namely the basal diet group (CON group), the FDG replacing 10% concentrate (FDG-10%) group, and the FDG replacing 20% concentrate (FDG-20%) group. Each group of 10 cattle was housed in three different pens, each with a feeder and drinking trough. The basal diet was formulated with reference to the nutritional requirements of the Chinese "Beef Cattle Feeding Standard" (NY/T 815-2004) for a body weight of 300 kg and an ADG of 1 kg/day. The ratio of forage to concentrate was 45:55, and the dietary composition of each group is shown in Table S1. Finally, a 30-day formal feeding period was conducted according to the formulation.

## Sample collection and processing

At the end of the formal experimental period (30th day of the formal feeding), before morning feeding, fecal samples were collected from six cattle randomly selected from the CON, FDG-10%, and FDG-20% groups. Feces were collected by rectal stimulation and transferred into 2 mL freezing tubes. The tubes were snap-frozen in liquid nitrogen immediately and then stored at −80°C for the following determination of fecal bacterial and fungal community compositions and metabolomes.

## High-throughput sequencing and analysis of 16S rDNA and ITS

Genomic DNA from the samples was extracted using a Fecal Genomic DNA Extraction Kit (DP712), and the genomic DNA was used as a template to amplify the bacterial 16S rDNA V3-V4 highly variable region and the fungal ITS1 region using corresponding primers (16S rDNA V3-V4: 338F: ACTCCTACGGGAGGCAGCA and 806R: GGACTACHVGGGTWTC-TAAT; ITS1: 1737F: GGAAGTAAAAGTCGTAACAAGG and 2043R: GCTGCGTTCTTCATCGATG C). DNA purity and concentration were checked by 1% agarose gel electrophoresis. PCR products were mixed in aliquots according to their concentration and purified by electrophoresis on 1× TAE, 2% agarose gel using Qiagen's Gel Extraction Kit to recover the target bands. The TruSeq DNA PCR-Free Sample Preparation Kit was used for library construction, and after Qubit quantification and library testing, sequencing was performed on the Illumina NovaSeq 6000 sequencer with PE250 read lengths. The data for each sample were distinguished according to the barcode sequence, and the extracted data were stored in fastq format, and the PE data had two files, fq1 and fq2, for each sample, which were the reads at both ends of the sequencing. Valid data (clean data) were obtained after double-ended sequence splicing using FLASH software (http://ccb.jhu.edu/software/FLASH/), along with quality control filtering for read quality and the effect of merging. To investigate the diversity of the species composition of the samples, the clean reads of all samples were clustered using Uparse (http://drive5.com/uparse/), and the sequences were clustered into OTUs with 97% consistency, and then the representative sequences of the OTUs were compared with the corresponding reference data for species annotation using the RDP classifier algorithm. The representative sequences of the OTUs were then matched to the corresponding reference data for species annotation using the RDP classifier algorithm. The database used for 16S annotation was Silva 132 (https://www.arb-silva.de/) and the database

used for ITS annotation was UNITE (https://unite.ut.ee/). Alpha diversity, including Chao1, Shannon, and Simpson indices, was used to determine the abundance and diversity of the bacterial community, and dilution curves corresponding to the indices were generated to assess the saturation of microbial community detections for the entire experiment. Dynamic visualization of individual sample species annotation results and abundance using Krona software. Analyses such as Venn, barplot, and heatmap plots to compare species composition between samples or subgroups. PCoA downscaling analyses to compare similarities and differences in species composition between samples or subgroups. A LEfSe approach was used to identify the categorical traits that differed most at each level based on the Kruskal-Wallis (KW) rank sum test, with the significance threshold for the KW test set at 0.05. The cut-off score for log-linear discriminant analysis (LDA) was set at 2.0.

## LC/MS-based metabonomic determination and data analysis

The LC-MS metabolomics was used to detect metabolites in the feces of the CON, FDG-10%, and FDG-20% groups. Briefly, the samples were slowly thawed at 4°C and added to pre-cooled methanol/acetonitrile/water solution (2:2:1, vol/vol), vortexed and mixed, sonicated at low temperature for 30 min, stood at −20°C for 10 min, centrifuged at 14,000 × $g$ at 4°C for 20 min, and the supernatant was vacuum dried. After the samples were slowly thawed at 4°C, an appropriate amount of samples was added to pre-cooled methanol/acetonitrile/water solution (2:2:1, vol/vol), vortexed and mixed, and then sonicated at low temperature for 30 min, and then allowed to stand at −20°C for 10 min, and then centrifuged at 14,000 × $g$ for 20 min at 4°C, and the supernatant was dried under vacuum, and then added to 100 µL of acetonitrile aqueous solution (acetonitrile: water = 1:1, vol/vol) for redissolution during mass spectrometry (MS) analysis. Separation was performed using a Vanquish LC ultra-high performance liquid chromatography (UHPLC) HILIC column. QC samples were included in the sample queue to monitor and evaluate the stability of the system and the reliability of the experimental data. After separation on the Vanquish LC UHPLC system, the samples were analyzed by mass spectrometry on a Q Exactive series mass spectrometer (Thermo) and detected by electrospray ionization (ESI) in positive and negative ion modes. The ESI source and mass spectrometry were set up with the following parameters: nebulizing gas, auxiliary heating gas 1 (Gas1): 60, auxiliary heating gas 2 (Gas2): 60, curtain gas (CUR): 30 psi, ion source temperature: 600°C, and spray voltage (ISVF) ±5,500 V (both positive and negative modes). Raw data were converted to mzXML format using ProteoWizard, and then XCMS software was used for peak alignment, retention time correction, and peak area extraction. The data obtained by XCMS extraction were first subjected to metabolite structure identification and data pre-processing, then to experimental data quality assessment, and finally to data analysis. Multidimensional statistical analyses, including unsupervised PCA) and OPLS-DA, were performed on the data from the CON group vs FDG-10% group and CON group vs FDG-20% group, respectively. Combined with the importance of variables projected in OPLS-DA (VIP) and $P$ values obtained in the Student's $t$ test analysis, the differential metabolites (VIP > 1, $P < 0.05$) were selected, and the fold change (FC) of each metabolite was calculated by comparing the mean values between groups. KEGG functional annotation and enrichment analyses of the differential metabolites were performed.

## Correlation analysis

Correlation coefficients between significantly different flora (LEfSe LDA >2 and $P < 0.05$) and significantly different metabolites (VIP > 1 and $P < 0.05$) were analyzed using Spearman statistics, which were carried out by combining the R language and for matrix heat map, hierarchical clustering, correlation network, etc. to explore the interactions between flora and metabolites from multiple perspectives.

## ACKNOWLEDGMENTS

This study was supported by the project of the Guizhou Provincial Department of Agriculture and Rural Affairs (No. [2022]163), the Guizhou Province Rural Industrial Revolution Cattle and Sheep Industry Development (Z20210001), the Guizhou University Talent Introduction Research Project (GDRJHZ[2020]63), the Guizhou University Cultivation Project (GDPY [2020]85), the Guizhou Science and Technology Plan Project (QKHJC-ZK [2022]YB158), and the Guizhou Science and Technology Support Fund Project (No. [2021]5646).

R.Z., E.Z., and C.C. conceived the study; R.Z., G.H., Z.C., L.C., and S.M. performed the experiments; S.M., G.H., M.Z., B.Z., K.W., E.Z., and C.C. analyzed experimental results and data; Z.C., B.Z., C.W., E.Z., and C.C. assisted with the animal experiments; R.Z. wrote the manuscript. All authors have read and agreed to the published version of the manuscript.

## AUTHOR AFFILIATIONS

[1]College of Animal Science, Guizhou University, Guiyang, China
[2]Guizhou Provincial Animal Disease Research Laboratory, Guiyang, China

## AUTHOR ORCIDs

Rong Zhang  http://orcid.org/0009-0009-5910-6241
Erpeng Zhu  http://orcid.org/0000-0003-2537-0801
Chao Chen  http://orcid.org/0009-0001-3615-7830

## DATA AVAILABILITY

Rong Zhang, Shihui Mei, Guangxia He, Miaozhan Wei, Lan Chen, Ze Chen, Min Zhu, Bijun Zhou, Kaigong Wang, Zhentao Cheng, Chunmei Wang, Erpeng Zhu, Chao Chen. 2024. Data from "Multi-omics analyses reveal fecal microbial community and metabolic alterations in finishing cattle fed probiotics-fermented distiller's grains diets" China National GeneBank Sequence Archive (CNSA) of the China National GeneBank DataBase (CNGBdb) (accession no. CNP0006074).

## ETHICS APPROVAL

Protocols for this experiment were reviewed and approved by the Animal Ethics Committee of Guizhou University (Guizhou, China) (approval number: EAE-GZU-2022-E039) and were under the academy's guidelines for animal research.

## ADDITIONAL FILES

The following material is available online.

### Supplemental Material

**Supplemental Material (Spectrum00721-24-s0001.docx).** Supplemental tables and figures.

### Open Peer Review

**PEER REVIEW HISTORY (review-history.pdf).** An accounting of the reviewer comments and feedback.

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
