## [Reviewer comments · Microbiology Spectrum]

Microbiology Spectrum

Multimomics analyses reveal fecal microbial community and metabolic alterations in finishing cattle fed probiotics-fermented distiller's grains diets

Rong Zhang, Shihui Mei, Guangxia He, Miaoqian Wei, Lan Chen, Ze Chen, Min Zhu, Bijun Zhou, Kaigong Wang, Zhentao Cheng, Chunmei Wang, Erpeng Zhu, and Chao Chen

Corresponding Author(s): Erpeng Zhu, Guizhou University College of Animal Science

Review Timeline:

Submission Date:	March 23, 2024
Editorial Decision:	August 4, 2024
Revision Received:	August 20, 2024
Editorial Decision:	November 24, 2024
Revision Received:	December 10, 2024
Accepted:	March 13, 2025

Editor: Francisco Uzal

Reviewer(s): Disclosure of reviewer identity is with reference to reviewer comments included in decision letter(s). The following individuals involved in review of your submission have agreed to reveal their identity: Adetomiwa Ayodele Adeniji (Reviewer #1); Ritesh Kumar Aggarwal (Reviewer #2); Yuchao Zhao (Reviewer #3)

Transaction Report:

DOI: <https://doi.org/10.1128/spectrum.00721-24>

Re: Spectrum00721-24 (Multiomics analyses reveal intestinal microbial community and metabolic alterations in finishing cattle fed probiotics-fermented distiller's grains diets)

Dear Dr. Erpeng Zhu:

Thank you for the privilege of reviewing your work. Below you will find my comments, instructions from the Spectrum editorial office, and the reviewer comments.

Revision Guidelines

Sincerely,
Francisco Uzal
Editor
Microbiology Spectrum

Reviewer #1 (Comments for the Author):

This study investigates the effects of probiotics-fermented distiller's grains (FDG) diets on the intestinal microbial community and metabolic profiles of finishing cattle.

The study elucidates the potential benefits of feeding FDG-based diets to finishing cattle.

Significant findings in the study reveal that feeding FDG-based diets to finishing cattle: 1. improved their intestinal metabolite; 2. enhanced bile secretion; 3. could bring potential improvement in taste transduction; 4. could help maintain the intestinal microbial community.

The manuscript is well presented.

Reviewer #2 (Comments for the Author):

These authors have written a good manuscript with precise findings presented as such. I found the writing style rather interesting and easy to follow. However there are a few inconsistencies in numbers and data presented. I have highlighted a couple of patterns and marked some highlights in the available PDF file. The text for my comments is as below.

Authors need to use the full form of OTUs at the first instance of use. If possible, also define why OTU is chosen.

The results section and each figure legend needs to open with the remark on what the results show. For example - the figure legend for Fig 1 could say that Bacterial diversity across the three groups is relatively stable/similar in the opening sentence. And Fig 2 can open by saying that the dominant phyla were Bacteroidetes and Firmicutes at the phylum level. At the genus level, Ruminococcaceae.UCG.005 and Rikenellaceae.RC9.gut.group were the dominant group.

Figure 1: "Figure legend" has spelling error - says Figure. (A) Numbers do not match the Results and Figure - Venn diagrams show 756, 1189 and 1197 while the results show 1689, 1639, 1464. Please correct or make it clear how certain overlaps/unions in Venn diagrams add. (B) Group the CON, FDG10 and FDG20 curves to increase legibility. Figure 1C in text is referred to after Figure 1D; not a chronological order of mentions in the text.

**Multiomics analyses reveal intestinal microbial community and metabolic**
**alterations in finishing cattle fed probiotics-fermented distiller's grains diets**

Rong Zhang^{a,b}, Shihui Mei^{a,b}, Guangxia He^{a,b}, Miaozhan Wei^{a,b}, Lan Chen^{a,b}, Ze Chen^{a,b},
Min Zhu^{1,2}, Bijun Zhou^{a,b}, Kaigong Wang^{a,b}, Zhentao Cheng^{a,b}, Chunmei Wang^a, Erpeng
Zhu^{a,b,#}, Chao Chen^{a,#}

a. College of Animal Science, Guizhou University, Guiyang 550025, China

b. Guizhou Provincial Animal Disease Research Laboratory, Guiyang 550025, China

**Corresponding author's email address**

#Address correspondence to Erpeng Zhu, zhu13782701756@126.com; Chao Chen,
chenc@gzu.edu.cn

**Abstract**

Distiller's grains (DG) are a potential source of animal feeds, and many studies have
indicated positively regulatory roles of feeding DG diets in animal breeding. However,
there is currently a dearth of research on the actions and underlying mechanisms of
probiotics-fermented distiller's grains (FDG)-based diets in cattle breeding. This study
aimed to assess the impact of integrating FDG into diet of finishing cattle on their
intestinal microbial community and metabolites. Thirty Simmental crossbred cattle (local

yellow cattle × Simmental cattle, 6.5 months old, 265.0 ± 2.48 kg) were selected and
randomly divided into three dietary treatments, including the basal diet group (CON
group), the FDG replacing 10% concentrate (FDG-10%) group, and the FDG replacing
20% concentrate (FDG-20%) group. 16S and ITS sequencing of fecal samples collected
from each group at 45 days suggested that feeding FDG diets had little effect on the
composition and diversity of intestinal bacterial and fungal communities in finishing
cattle. However, the relative abundance of cellulose-degrading bacteria including the
*Christensenellaceae R-7* group and *Ruminococcaceae* family was significantly higher in
both the FDG-20% vs CON comparison and the FDG-20% vs FDG-10% comparison.
Besides, the FDG-10% group had a significant drop in relative abundance of *Aspergillus*
and a noteworthy increase in relative abundance of *Candida*, when compared to CON
group. Non-targeted metabolomic analysis showed that addition of FDG modified the
levels of Organoheterocyclic compounds, Lipids and lipid-like molecules, and
Benzenoids in feces of finishing cattle and significantly enhanced the metabolic pathway
of Bile secretion. Further correlation analyses suggested a close association between the
significantly differential intestinal microflora and metabolites. In conclusion, these results
suggest that FDG supplementation has little effect on the structure and diversity of the
intestinal microbiota in finishing cattle, but alters intestinal metabolite profiles and
influences bile secretion pathways by modulating the relative abundance of genus of
intestinal bacteria and fungi *Christensenellaceae R-7* group,

*Lachnospiraceae_NK3A20_group, Mucor, Candida*. These findings provide a scientific
theoretical basis for the use of FDG in animal feeds.

**Importance**

FDG are potential feed source for livestock, here we investigated the effects of partially
replacing concentrates with FDG on fecal bacterial and fungal community structure and
metabolic profiles in finishing cattle. The results reveal that feeding FDG-based diets
alter intestinal metabolite profiles and up-regulate bile secretion pathway through the
regulation of relative abundance of certain intestinal genera. These findings provide some
new insights into clarifying the role and potential mechanisms of FDG diets and also
offer a scientific basis for development of DG into functional feed resources.

**Keywords:** Probiotics-fermented distiller's grains, feces, metabolomics, intestinal
microbiota, ITS

**1 . Introduction**

With the rapid development of the global livestock industry, issues of feed shortages and
increasing prices have arisen. To produce safe, inexpensive and nutritious feed has
become one of the most urgent issues in livestock husbandry. Distiller's grains (DG), a

by-product of the wine industry, are favored because of their low production cost and
high nutritional value. Studies have shown that DG have good effects as an alternative
feed for different animal species (1). One of the primary uses of DG is to replace a certain
proportion of concentrates to feed livestock, including direct feeding and feeding after
drying or fermentation treatment, of which microbial fermentation of DG has a wider
range of applications and gives better results. Microbial fermentation mainly adopts
probiotic fermentation. Probiotics-fermented distiller's grains (hereafter collectively
referred to as fermented distiller's grains, FDG) is mainly utilized due to the fact that
probiotics can use the sugars and starch present in DG to ferment and produce a variety
of nutrients such as proteins, cellulase, fatty acids, etc. (3). Studies have also shown that
FDG can improve the quality and palatability of DG, inhibit the proliferation of harmful
bacteria and maintain the micro-ecological balance of the intestinal tract in Simmental
crossbred cattle and broiler (4). Therefore, FDG has become one of the hot topics of
research into the feed utilization of DG.

The development of high-throughput sequencing technology, high-resolution mass
spectrometry, and data integration and analysis technology has fostered new
breakthroughs in systems biology research characterized by multi-omics (6). Multi-omics
aims to integrate genomics, epigenomics, transcriptomics, proteomics and metabolomics
in an unbiased way to systematically resolve complex mechanisms and phenotypes of
animal life systems. Intestinal microbes and their interactions with the host play a crucial

**role** in the health of the host organism (7). Multi-omics can be used to systematically
study the biology of the host intestine and to characterize host-intestinal microbiome
interactions using deeply integrated technologies, and this can reveal the complex
regulatory mechanisms associated with the growth and development of animals, as well
as the mechanisms underlying the development and treatment of various diseases (8). In
recent years, omics analysis has also been used to study the effects of FDG diets on
animals. Our previous studies have suggested that feeding dried distillers' grains and
FDG fermented by commercial microbial preparations altered the microbial community
structure and metabolic patterns of gastrointestinal tract in Guanling cattle and Guanling
crossbred cattle, and that replacing part of the concentrates with FDG fermented by
commercial microbial preparations also altered the amino acid composition and
proportions in beef, thus **improving its flavor in** Guanling cattle (11). To further
investigate the effects of FDG diets on intestinal microbial structure and metabolic
profiles, here we selected four widely used probiotics (*Enterococcus faecalis*,
*Lactobacillus plantarum*, *Aspergillus niger*, and *Saccharomyces cerevisiae*) to
supplement FDG and then partially replaced the concentrates of basal diets to feed
finishing cattle. Finally, 16S and ITS high-throughput sequencing and liquid
chromatography-mass spectrometry (LC-MS) metabolomics were performed to
investigate the effects of feeding FDG diets on the structure of the intestinal microflora
and the metabolome of finishing cattle, with the aim of providing reference for the

application of FDG as feed additives in livestock husbandry.

**2 . Results**

**2.1 Effect of FDG diets on the structure of intestinal bacterial communities in**

**finishing cattle**

[revised manuscript text omitted]

Korosec A, Brown M, Vaahromeri K, Duggan M, Kerjaschki D, Esterbauer H,
Colinge J, Eisenbarth SC, Decker T, Bennett KL, Kubicek S, Sixt M, Superti-Furga
G, Knapp S. 2016. Heme drives hemolysis-induced susceptibility to infection via
disruption of phagocyte functions. *Nat Immunol* 17:1361-1372.
- (38) Koopman N, Katsavelis D, Hove AST, Brul S, Jonge WJ, Seppen J. 2021. The
multifaceted role of serotonin in intestinal homeostasis. *Int J Mol Sci* 22:9487-9509.
- (39) Li TG, Apte U. 2015. Bile acid metabolism and signaling in cholestasis,
inflammation, and cancer. *Adv Pharmacol* 74:263-302.
- (40) Ridlon JM, Kang DJ, Hylemon PB. 2006. Bile salt biotransformations by human
intestinal bacteria. *J Lipid Res* 47:241-59.
- (41) Kriaa A, Bourgin M, Potiron A, Mkaouar H, Jablaoui A, Gérard P, Maguin E, Rhimi
656 M. 2019. Microbial impact on cholesterol and bile acid metabolism: current status
and future prospects. *J Lipid Res* 60:323-332.
- (42) Collins SL, Stine JG, Bisanz JE, Okafor CD, Patterson AD. 2023. Bile acids and the
gut microbiota: metabolic interactions and impacts on disease. *Nat Rev Microbiol*

21:236-247.

(43) Kitamura A, Tsurugizawa T, Uematsu A, Uneyama H. 2014. The sense of taste in the
upper gastrointestinal tract. *Curr Pharm Des* 20:2713-2724.

(44) Raybould HE. 2010. Gut chemosensing: interactions between gut endocrine cells and
visceral afferents. *Auton Neurosci* 153:41-46.

(45) Haber AL, Biton M, Rogel N, Herbst RH, Shekhar K, Smillie C, Burgin G, Delorey
TM, Howitt MR, Katz Y, Tirosh I, Beyaz S, Dionne D, Zhang M, Raychowdhury R,
Garrett WS, Rozenblatt-Rosen O, Shi HN, Yilmaz O, Xavier RJ, Regev A. 2017. A
single-cell survey of the small intestinal epithelium. *Nature* 551:333-339.

(46) Kaelberer MM, Buchanan KL, Klein ME, Barth BB, Montoya MM, Shen X, Bohó
rquez DV. 2018. A gut-brain neural circuit for nutrient sensory transduction. *Science*
361:e5236-e5253.

**Finure legend**

**Fig. 1 Analysis of Intestinal Bacterial Community Diversity. (A) Venn diagram**

illustrating the operational taxonomic units (OTUs) in three groups. (B) Rarefaction

curve. (C) Principal-coordinate analysis. (D) Alpha diversity analysis. The basal diet

group, the FDG replacing 10% concentrate group, and the FDG replacing 20%

concentrate group are denoted as CON, FDG-10%, and FDG-20%, respectively. Each

experimental group comprises fecal samples randomly collected from four cattle

(n=4).

Fig. 2 Analysis of Intestinal Bacterial Community Structure. (A) Proportions of
 bacterial communities by phylum. (B) Proportions of bacterial communities by genus.
 (C-E) The Linear Discriminant Analysis Effect Size (LEfSe) analyses indicate
 differences in intestinal bacterial community composition between groups, with a
 Linear Discriminant Analysis (LDA) score of > 2 and P -value of < 0.05 . The basal
 diet group, the FDG replacing 10% concentrate group, and the FDG replacing 20%
 concentrate group are denoted as CON, FDG-10%, and FDG-20%, respectively. Each
 experimental group comprises fecal samples randomly collected from four cattle
 ($n=4$). Abbreviations used were: p for phylum, c for class, o for order, f for family,
 and g for genus.

Fig. 3 Analysis of Intestinal Fungal Community Diversity. (A) Venn diagram

illustrating the operational taxonomic units (OTUs) in three groups. (B) Rarefaction

curve. (C) Principal-coordinate analysis. (D) Alpha diversity analysis. The basal diet

group, the FDG replacing 10% concentrate group, and the FDG replacing 20%

[revised manuscript text omitted]

These authors have written a good manuscript with precise findings presented as such. I found the writing style rather interesting and easy to follow. However there are a few inconsistencies in numbers and data presented. I have highlighted a couple of patterns and marked some highlights in the available PDF file. The text for my comments is as below.

Authors need to use the full form of OTUs at the first instance of use. If possible, also define why OTU is chosen.

The results section and each figure legend needs to open with the remark on what the results show. For example - the figure legend for Fig 1 could say that Bacterial diversity across the three groups is relatively stable/similar in the opening sentence. And Fig 2 can open by saying that the dominant phyla were Bacteroidetes and Firmicutes at the phylum level. At the genus level, Ruminococcaceae.UCG.005 and Rikenellaceae.RC9.gut.group were the dominant group.

Figure 1: "Figure legend" has spelling error – says Figure. (A) Numbers do not match the Results and Figure – Venn diagrams show 756, 1189 and 1197 while the results show 1689, 1639, 1464. Please correct or make it clear how certain overlaps/unions in Venn diagrams add. (B) Group the CON, FDG10 and FDG20 curves to increase legibility. Figure 1C in text is referred to after Figure 1D; not a chronological order of mentions in the text.

Response to Reviewer #1 Comments

Reviewer #1 (Comments for the Author):

This study investigates the effects of probiotics-fermented distiller's grains (FDG) diets on the intestinal microbial community and metabolic profiles of finishing cattle.

The study elucidates the potential benefits of feeding FDG-based diets to finishing cattle.

Significant findings in the study reveal that feeding FDG-based diets to finishing cattle: 1. improved their intestinal metabolite; 2. enhanced bile secretion; 3. could bring potential improvement in taste transduction; 4. could help maintain the intestinal microbial community.

The manuscript is well presented.

Response: Thank you very much for your professional comments.

Response to Reviewer #2 Comments

Reviewer #2 (Comments for the Author):

These authors have written a good manuscript with precise findings presented as such. I found the writing style rather interesting and easy to follow. However there are a few inconsistencies in numbers and data presented. I have highlighted a couple of patterns and marked some highlights in the available PDF file. The text for my comments is as below.

Point 1: Authors need to use the full form of OTUs at the first instance of use. If possible, also define why OTU is chosen.

Response 1: Thank you very much for your comments. We have added the full form of OTUs at the first instance of use (**Line 104**). And we also explained why OTUs were chosen in the text as follows: “Operational taxonomic units (OTUs) are taxonomic unit markers in phylogenetic and population genetics studies. Generally, sequences with 97% similarity were assigned to the same OTUs, which means a microbial species or genus. The diversity and the abundance of different microbes in a test sample are based on the analysis of OTUs.” (**Line 104-108**). The revised part is marked in red in the revision. We wish that the changes will meet with approval.

Point 2: The results section and each figure legend needs to open with the remark on what the results show. For example - the figure legend for Fig 1 could say that Bacterial diversity across the three groups is relatively stable/similar in the opening sentence. And Fig 2 can open by saying that the dominant phyla were Bacteroidetes and Firmicutes at the phylum level. At the genus level, Ruminococcaceae.UCG.005 and Rikenellaceae.RC9.gut.group were the dominant group.

Response 2: Your suggestion is valuable. We have modified the results section and each figure legend as you suggest (**Line 102, 145, 176, 216, 681, 688, 700, 707, 717, 725, 735, and 742**). The revised part is marked in red in the revision.

Point 3: Figure 1: "Figure legend" has spelling error - says Finure.

Response 3: “Finure” was changed to “Figure”. (**Line 680**)

Point 4: (A) Numbers do not match the Results and Figure - Venn diagrams show 756, 1189 and 1197 while the results show 1689, 1639, 1464.

Response 4: We feel sorry for our mistakes. We have checked and modified the results to be consistent with the figure as follows: “A Venn diagram showed that we identified 2380 common OTUs in all samples analyzed, with 1670, 1655 and 1439 OTUs specific to the CON, FDG-10%, and FDG-20% group, respectively.” (**Line 109**)

Point 5: (B) Group the CON, FDG10 and FDG20 curves to increase legibility.

Response 5: Your suggestion is valuable. We have grouped Group the CON, FDG10 and FDG20 curves (Fig. 1B and Fig. 3B) by using different colors to increase legibility of the figure.

Point 6: Figure 1C in text is referred to after Figure 1D; not a chronological order of mentions in the text.

Response 6: Your suggestion is valuable. We have checked and modified the numbering order of Figure 1C and 1D in the Figure 1 and Figure 3 to keep the chronological order of mentions in the text.

We have read all comments carefully, which are very important and helpful for modifying and improving our manuscript, and we have made corrections or explanations accordingly which we hope meet with approval. We appreciated for your warm work earnestly and sincerely hope that the correction will meet with approval. Once again, thank you very much.

Best regards!

Yours sincerely,

Dr. Er-Peng Zhu and Chao Chen on behalf of all the authors

College of Animal Science, Guizhou University, Guiyang 550025, China.

E-mail: zhu13782701756@126.com; chenc@gzu.edu.cn

Re: Spectrum00721-24R1 (Multiomics analyses reveal intestinal microbial community and metabolic alterations in finishing cattle fed probiotics-fermented distiller's grains diets)

Dear Dr. Erpeng Zhu:

Thank you for the privilege of reviewing your work. Below you will find my comments, instructions from the Spectrum editorial office, and the reviewer comments.

Revision Guidelines

Sincerely,
Francisco Uzal
Editor
Microbiology Spectrum

Reviewer #3 (Comments for the Author):

The authors investigated the effects of probiotic-fermented distiller's grains (FDG) diets on the fecal microbial community and metabolic profiles of finishing cattle. Although some revisions have been made, several important issues remain that require clarification. Specifically, as fecal samples were collected, the term "intestinal microbiota" is inappropriate. Additionally, several critical phenotypic data are lacking, such as production performance, fecal VFA concentrations, and blood biochemical

indicators.

In the Importance section, please spell out FDG in full.

In lines 60-61, the authors refer to "studies" but only cite a single reference. Please add additional citations.

In lines 98-99, the authors mention "and the metabolome of finishing cattle, with the aim of providing a reference for the application of FDG as feed additives in livestock husbandry." Please be more specific: what references or guidance does this study intend to provide?

In line 106, there is a typographical error; it should read "OTU" instead of "OUT." Please check the entire manuscript for similar errors.

Since fecal samples were collected, it would be more appropriate to use the term "fecal microbiota" rather than "intestinal microbiota."

In lines 417-418, the authors state that "the dietary composition of each group is shown in Table S1. Finally, a 45-day feeding trial was conducted according to the formulation." The trial period is stated as 45 days, but what was the adaptation period? Production performance data are essential and were not reported in this paper. Please provide growth data for the finishing cattle.

Why was the rumen microbiota not studied in favor of the hindgut microbial community? If fecal samples are the focus, why were VFA concentrations not measured? These data should be included.

The authors only performed correlation analyses between microbiota and metabolites. Without phenotypic data, what is the significance of these correlations?

Response to Reviewer #3 Comments

Reviewer #3 (Comments for the Author):

The authors investigated the effects of probiotic-fermented distiller's grains (FDG) diets on the fecal microbial community and metabolic profiles of finishing cattle. Although some revisions have been made, several important issues remain that require clarification. Specifically, as fecal samples were collected, the term "intestinal microbiota" is inappropriate.

Additionally, several critical phenotypic data are lacking, such as production performance, fecal VFA concentrations, and blood biochemical indicators.

Point 1: In the Importance section, please spell out FDG in full.

Response 1: Thank you very much for your comments. We have added the full form of FDG in the Importance section. (**Line 44**)

Point 2: In lines 60-61, the authors refer to "studies" but only cite a single reference. Please add additional citations.

Response 2: Thank you very much for your comments. We feel sorry for our unrigorous expression and have added relevant citations (**Line 62**). The inserted references were also listed as follows:

References:

(3) Ding XM, Qi YY, Zhang KY, Tian G, Bai SP, Wang JP, Peng HW, Lv L, Xuan Y, Zeng QF. 2022 Corn distiller's dried grains with solubles as an alternative ingredient to corn and soybean meal in Pekin duck diets based on its predicted AME and the evaluated standardized ileal digestibility of amino acids. *Poult Sci* 101(8):101974.

(4) Spinler MS, Tolosa AF, Gebhardt JT, Tokach MD, Goodband RD, DeRouchey JM, Coble KF, Woodworth JC. 2023. Comparing tryptophan:lysine ratios in dried distillers grains with solubles-based diets with and without a dried distillers grains with solubles withdrawal strategy on growth, carcass characteristics, and carcass fat iodine value of growing-finishing pigs. *J Anim Sci* 101:skad245.

(5) Clizer DA, Tostenson BJ, Frederick B, Cline PM, Samuel RS. 2023. Performance response of increasing the standardized ileal digestible tryptophan:lysine ratio in diets containing 40% dried distiller grains with solubles. *J Anim Sci* 101:skad264.

Point 3: In lines 98-99, the authors mention "and the metabolome of finishing cattle, with the aim of providing a reference for the application of FDG as feed additives in livestock husbandry." Please be more specific: what references or guidance does this study intend to provide?

Response 3: Your suggestion is very valuable. We have accordingly modified these sentences as follows: "Finally, 16S and ITS high-throughput sequencing and liquid chromatography-mass spectrometry (LC-MS) metabolomics were performed to investigate

the effects of feeding FDG diets on the structure of the fecal microbiota and the metabolome of finishing cattle, thus screening out the key flora, metabolites, and metabolic pathways as potential biomarkers, which will provide references for the feasibility of FDG as an alternative animal feed and shed new light on the mitigation of the shortage of feed resources in the livestock industry”. (Line 99-102)

Point 4: In line 106, there is a typographical error; it should read "OTU" instead of "OUT." Please check the entire manuscript for similar errors.

Response 4: We feel sorry for our mistakes. “OUT” was changed to “OTU” (Line 111). We have also checked the entire manuscript for similar errors: “stabilises” was changed to “stabilizes” (Line 149); “was” was changed to “were” (Line 214).

Point 5: Since fecal samples were collected, it would be more appropriate to use the term "fecal microbiota" rather than "intestinal microbiota."

Response 5: Your suggestion is of great value. We have checked the entire manuscript and replaced the term "intestinal microbiota" with "fecal microbiota" if applicable (Lines 1, 19, and 26, etc.).

Point 6: In lines 417-418, the authors state that "the dietary composition of each group is shown in Table S1. Finally, a 45-day feeding trial was conducted according to the formulation." The trial period is stated as 45 days, but what was the adaptation period?

Response 6: Thank you very much for your comments. We feel sorry for our unclear expressions and have re-described relevant sentences in detail in the revision: “The trial consisted of a 15-day adaptation period and a 30-day formal feeding period” (Line 412); “Finally, a 30-day formal feeding period was conducted according to the formulation” (Line 421).

Point 7: Production performance data are essential and were not reported in this paper. Please provide growth data for the finishing cattle.

Response 7: Thank you very much for your valuable comments. Since there was no significant change ($P > 0.05$) in the average daily gain (ADG) of finishing cattle after feeding FDG diets, we did not present this data in the manuscript. We have included this data as supplementary material in the **Table S2**, which is also listed as follows for your reviewing.

Table S2 Effects of dietary FDG on the average daily gain (ADG) of finishing cattle

Item	Groups			SEM	P-Value
	CON	FDG-10%	FDG-20%		
ADG, Kg	0.92±0.04 ^a	0.96±0.05 ^a	0.87±0.05 ^a	0.04	0.75

ADG: average daily gain.

CON, FDG-10%, and FDG-20% represent the group without FDG supplementation, the group with 10% FDG substituting for 10% concentrate, and the group with 20% FDG substituting for 20% concentrate, respectively.

Data are presented as means \pm SEM (n=6), and the same row with same superscript indicates no significant difference ($P > 0.05$).

Point 8: Why was the rumen microbiota not studied in favor of the hindgut microbial community? If fecal samples are the focus, why were VFA concentrations not measured? These data should be included.

Response 8: Thank you very much for your professional comments. In fact, we have conducted a systematic study concurrently, encompassing the bacteriome, fungiome and metabolome of oral, rumen, and fecal samples, as well as rumen fermentation parameters, serum immunological and biochemical indices. We intended to systematically investigate the impact of feeding FDG diets on the microbiota and metabolism of the entire digestive tract in finishing cattle. Because the results of the oral and rumen samples, as well as rumen fermentation parameters and blood immunological and biochemical indices, have been submitted to another journal for reviewing (current status: under review). Therefore, these data cannot be presented in this manuscript. We have provided some of the relevant data for reviewing here (as shown below). VFAs are important intermediates in the anaerobic digestion process and play important roles in maintaining the integrity of the intestinal barrier, regulating the chemotaxis of immune cells, and anti-bacterial and anti-inflammatory effects. VFA assays will make our results more comprehensive and compelling. As for the VFA concentrations not being measured, we admit that it is our mistake during the experimental design, and the measurement of the VFA concentrations will be included in our plan for the following studies. Thank you very much for your understanding.

Effect of feeding FDG diets on rumen enzyme activities in finishing cattle. (A-I) The activities of β -glucosidase (β -GC), protease, amylase, pectinase, filter paper cellulose (FPase), xylanase, endoglucanase, microcrystalline cellulose (MCC), and lipase in rumen fluids were determined by ELISA. The basal diet group, the FDG replacing 10% concentrate group, and the FDG replacing 20% concentrate group are denoted as CON, FDG-10%, and FDG-20%, respectively. Each experimental group comprises rumen fluid samples randomly collected from four cattle ($n=4$). Data are expressed as mean \pm SEM and significance is ns ($P > 0.05$), * ($P < 0.05$) and ** ($P < 0.01$), respectively.

Effect of feeding FDG diets on rumen bacterial community structure in finishing cattle. (A) Proportions of bacterial communities at the phylum level. (B) Proportions of bacterial communities at the genus level. (C-E) The LEfSe analyses indicate differences in rumen bacterial community composition between groups, with a Linear discriminant analysis (LDA) score of > 2 and P -value of < 0.05 . The basal diet group, the FDG replacing

10% concentrate group, and the FDG replacing 20% concentrate group are denoted as CON, FDG-10%, and FDG-20%, respectively. Each experimental group comprises rumen fluid samples randomly collected from four cattle (n=4).

Point 9: The authors only performed correlation analyses between microbiota and metabolites. Without phenotypic data, what is the significance of these correlations?

Response 9: Thank you very much for your professional comments. Correlation analysis of microbiota and metabolism with phenotypic indices contributes certainly to identifying the relationship between the key differential microbiota and metabolites for samples with phenotypic differences. However, since the phenotypic indices we examined (e.g., rumen fermentation parameters, serum immunological and biochemical indices) mentioned above are currently available to another journal for reviewing, they cannot be included in this manuscript. Moreover, as described in the Response 7, there was no significant change ($P > 0.05$) in the average daily gain (ADG) of finishing cattle after feeding FDG diets (Table S2), and therefore no correlation analysis was performed using this phenotypic indicator. Although the growth performance indicators presented in this manuscript are inadequate, we think that the correlation analyses between microbiota and metabolites remain meaningful to some extent. Gut microbiota has multiple metabolic regulation functions, and correlation analysis may also help us and reveal changes in fecal differential flora-mediated metabolic functions in finishing cattle after feeding FDG diets. In fact, in our study, we found that fecal flora *Christensenellaceae R-7 group*, *Lachnospiraceae_NK3A20_group*, *Mucor*, *Candida* showed positive correlations with most of the differential metabolites, which may be involved in bile secretion metabolic pathway and maintenance of energy balance in finishing cattle. The combined analysis of differential flora and metabolites may reveal a potential relationship between altered fecal flora and metabolites after feeding FDG diets, which may be related to some changes in physiological properties of finishing cattle and needs further confirmation. In addition, feeding FDG diets did not significantly reduce the ADG and induce the production of harmful fecal flora in finishing cattle, suggesting the feasibility of FDG as an alternative animal feed. This realizes the feed utilization of FDG and becomes a potential strategy to alleviate of the problem of feed scarcity in the livestock industry.

We have read all comments carefully, which are very important and helpful for modifying and improving our manuscript, and we have made corrections or explanations accordingly. We appreciated for your warm work earnestly and sincerely hope that the correction will meet with approval. Once again, thank you very much.

Best regards!

Yours sincerely,

Dr. Er-Peng Zhu and Chao Chen on behalf of all the authors

College of Animal Science, Guizhou University, Guiyang 550025, China.

E-mail: zhu13782701756@126.com; chenc@gzu.edu.cn

Re: Spectrum00721-24R2 (**Multimics analyses reveal fecal microbial community and metabolic alterations in finishing cattle fed probiotics-fermented distiller's grains diets**)

Dear Dr. Erpeng Zhu:

Your manuscript has been accepted, and I am forwarding it to the ASM production staff for publication. Your paper will first be checked to make sure all elements meet the technical requirements. ASM staff will contact you if anything needs to be revised before copyediting and production can begin. Otherwise, you will be notified when your proofs are ready to be viewed.

Sincerely,
Ruth Ann Luna
Editor
Microbiology Spectrum